# Glucose Metabolism and Metabolomic Changes in Response to Prolonged Fasting in Individuals with Obesity, Type 2 Diabetes and Non-Obese People—A Cohort Trial

**DOI:** 10.3390/nu15030511

**Published:** 2023-01-18

**Authors:** Norbert J. Tripolt, Sebastian J. Hofer, Peter N. Pferschy, Faisal Aziz, Sylvère Durand, Fanny Aprahamian, Nitharsshini Nirmalathasan, Mara Waltenstorfer, Tobias Eisenberg, Anna M. A. Obermayer, Regina Riedl, Harald Kojzar, Othmar Moser, Caren Sourij, Heiko Bugger, Abderrahim Oulhaj, Thomas R. Pieber, Matthias Zanker, Guido Kroemer, Frank Madeo, Harald Sourij

**Affiliations:** 1Interdisciplinary Metabolic Medicine Trials Unit, Division of Endocrinology and Diabetology, Medical University of Graz, 8010 Graz, Austria; 2Institute of Molecular Biosciences, NAWI Graz, University of Graz, Humboldtstraße 50, 8010 Graz, Austria; 3BioTechMed Graz, 8010 Graz, Austria; 4Field of Excellence BioHealth, University of Graz, 8010 Graz, Austria; 5Inserm U1138, Equipe Labellisée par la Ligue Contre le Cancer, Centre de Recherche des Cordeliers, Institut Universitaire de France, Sorbonne Université, Université de Paris, 75006 Paris, France; 6Metabolomics and Cell Biology Platforms, Institut Gustave Roussy, 94805 Villejuif, France; 7Center for Biomarker Research in Medicine (CBmed), 8010 Graz, Austria; 8Institute for Medical Informatics, Statistics and Documentation, Medical University of Graz, 8010 Graz, Austria; 9Department of Sport Science, Division of Exercise Physiology and Metabolism, University of Bayreuth, 95440 Bayreuth, Germany; 10Division of Cardiology, Medical University of Graz, 8010 Graz, Austria; 11Department of Epidemiology and Population Health, College of Medicine and Health Sciences, Khalifa University Abu Dhabi, Al-Ain P.O. Box 17666, United Arab Emirates; 12Division of Endocrinology and Diabetology, Medical University of Graz, 8010 Graz, Austria; 13Department of Biology, Institut du Cancer Paris CARPEM, Hôpital Européen Georges Pompidou, AP-HP, 75015 Paris, France

**Keywords:** metabolic flexibility, obesity, type 2 diabetes, fasting, glucose metabolism, OGTT, IVGTT

## Abstract

Metabolic regulation of glucose can be altered by fasting periods. We examined glucose metabolism and metabolomics profiles after 12 h and 36 h fasting in non-obese and obese participants and people with type 2 diabetes using oral glucose tolerance (OGTT) and intravenous glucose tolerance testing (IVGTT). Insulin sensitivity was estimated by established indices and mass spectrometric metabolomics was performed on fasting serum samples. Participants had a mean age of 43 ± 16 years (62% women). Fasting levels of glucose, insulin and C-peptide were significantly lower in all cohorts after 36 h compared to 12 h fasting (*p* < 0.05). In non-obese participants, glucose levels were significantly higher after 36 h compared to 12 h fasting at 120 min of OGTT (109 ± 31 mg/dL vs. 79 ± 18 mg/dL; *p* = 0.001) but insulin levels were lower after 36 h of fasting at 30 min of OGTT (41.2 ± 34.1 mU/L after 36 h vs. 56.1 ± 29.7 mU/L; *p* < 0.05). In contrast, no significant differences were observed in obese participants or people with diabetes. Insulin sensitivity improved in all cohorts after 36 h fasting. In line, metabolomics revealed subtle baseline differences and an attenuated metabolic response to fasting in obese participants and people with diabetes. Our data demonstrate an improved insulin sensitivity after 36 h of fasting with higher glucose variations and reduced early insulin response in non-obese people only.

## 1. Introduction

At divergence with the current lifestyle in the Westernized world, historically humans lived in an environment characterized by periods of fasting and feasting. For the fasting phase, humans have acquired tools to manage metabolic challenges to maintain cellular functions at minimized resting metabolic rate expenditure. Such physiological adaptations include lipolysis, fatty acid oxidation, ketogenesis, autophagy, minimal glucogenesis and decreased glucose oxidation [1]. These adaptions to changing energy demand and supply and the associated changes in fuel selection are known as ‘metabolic flexibility’ [2].

Within the plethora of calory restriction (CR) options, intermittent fasting (IF) has emerged as a well-tolerated and effective intervention [3]. Besides leading to a reduction in caloric intake, IF stimulates autophagic flux, enhances mitochondrial biogenesis and suppresses the mTOR pathway during the fasting period [4]. 

Moreover, animal studies suggested neuroprotective effects [5] and an improvement of learning and memory after prolonged fasting in mice [6].

Of note, previous studies have demonstrated that starvation, prolonged fasting or a low-carbohydrate diet result in impaired glucose tolerance in healthy participants [7,8,9]. During extended food restriction, glucose levels decrease, resulting in a release of the counterregulatory pancreatic hormone glucagon [10] and a reduction of both pancreatic insulin content [11] and circulating insulin levels in healthy individuals. Consequently, refeeding results in increased postprandial glucose excursions in otherwise normo-glycemic subjects [9].

When solely focusing on the increased postprandial glucose excursions, this observation might be regarded as potentially detrimental. However, it could also be a feature of metabolic flexibility in healthy subjects. Interestingly, previous mechanistic studies mainly investigated the impact of prolonged fasting on glucose tolerance in healthy individuals [7,8,9]. However, studies in obese people or those with established type 2 diabetes mellitus, where weight reduction and hence dietary interventions represent a cornerstone of metabolic management, are scarce.

The aim of our study was to elucidate glucometabolic adaptions following 12 and 36 h in healthy non-obese, obese participants and patients with type 2 diabetes. Here, we provide strong evidence in favor of the notion that metabolic flexibility is a privilege of the non-obese that is lost with obesity and type 2 diabetes.

## 2. Materials and Methods

### 2.1. Study Population

The study population included a total of 60 participants, 20 participants of whom were healthy and non-obese, 20 obese (BMI > 30 kg/m^2^) and 20 who had been diagnosed with type 2 diabetes. Participants were identified via Primary Care, Diabetes Outpatient Clinics, the Graz Diabetes Registry for Biomarker Research (GIRO) and adverts. Participants in the healthy non-obese cohort were included if they were 18 years and older, had a body mass index (BMI) between 20.0 and 27.0 kg/m^2^ and a fasting plasma glucose level lower than 110 mg/dL (without medication). Participants of the obese cohort had to have a BMI > 30.0 kg/m^2^ and a fasting plasma glucose level < 110 mg/dL (without medication). Individuals were excluded from both these cohorts if they had a history of type 1 or type 2 diabetes or established cardiovascular disease, were taking weight loss medications, were heavily drinking (more than 15 alcoholic drinks/week), or were taking any glucose lowering, lipid lowering or antihypertensive medication. Additionally, pregnant, or breast-feeding women and individuals on corticosteroids or antidepressants within 6 months prior to study initiation were excluded.

Patients in the third cohort were required to have established type 2 diabetes on either diet or a monotherapy or combination therapy of oral glucose lowering drugs [12]. The trial was approved by the Ethics Committee of the Medical University of Graz (30–238 ex 17/18) and was conducted at the Interdisciplinary Metabolic Medicine Trials Unit, Division of Endocrinology and Diabetology at the Medical University of Graz, Austria. All participants provided written informed consent before enrolling in the study.

### 2.2. Study Design

We conducted a prospective, mechanistic study. All study participants attended the clinical research center on 2 days separated from each other by at least 4 days. The study visits were carried out after 12 h fasting or 36 h fasting, respectively. During the fasting phase prior to the study visits, the participants were only allowed to consume calorie-free beverages such as water, flavored carbonated water, unsweetened black or green tea and coffee. Participants were requested to stop eating at 8:00 pm on the day before the 12 h fasting study visit or at 8:00 p.m. two days prior to the 36 h fasting study visit, respectively. Both study visits commenced at 7:30 a.m. with a physical examination by the study physician and measuring waist-to-hip ratio (WHR) and blood pressure. Patients with type 2 diabetes were asked to omit the intake of the glucose lowering medication before the oral glucose tolerance test (OGTT) and intravenous glucose tolerance test (IVGTT) on the visit day. Subsequently, the fasting blood sample was taken and the OGTT was started.

In each cohort, 10 people (the first 10 to agree to the sub-study) participated in two additional visits, again separated from each other by at least 4 days, where IVGTTs were performed, once after 12 h of fasting and once after 36 h of fasting. Study visits of the sub-study also started at 7:30 a.m. with physical examination and intravenous cannulation. The IVGTT was started at 8 a.m. and lasted 100 min.

### 2.3. Oral Glucose Tolerance Test (OGTT)

A standard gauge cannula was placed into a subcutaneous vein for blood sampling. To prevent blood clotting in the cannula, it was occasionally flushed with sterile normal saline. A pre-meal blood sample was taken (-5 min) and then all subjects were asked to drink the 75 g glucose solution (Glucoral^®^ 75 citron, Germania Pharmazeutika, Vienna) within a period of 2–4 min (time 0 min). The blood sampled at each time point was collected in a fluoride oxalate tube (1 mL) for plasma glucose measurement and into a serum tube for analyses of insulin and C-peptide.

#### 2.3.1. Homeostasis Model Assessment-Insulin Resistance

Homeostasis model assessment-insulin resistance (HOMA-IR) was first developed in 1985 by Matthews et al. and is a method used to quantify insulin resistance and beta-cell function from basal (fasting) glucose (G0) and fasting insulin (I0) concentrations [13]. The equation proposed by Matthews et al. is:HOMAIR = (I0 × G0)/22.5.

#### 2.3.2. Quantitative Insulin Sensitivity Check Index

The quantitative insulin sensitivity check index (QUICKI) is an empirically derived mathematical transformation of fasting blood glucose and plasma insulin concentrations that provides a consistent and precise insulin sensitivity index with a better positive predictive power. The QUICKI can be determined from fasting plasma glucose G0 (mg/dL) and fasting insulin I0 (µIU/mL) concentrations [14].
QUICKI = 1/(logI0 + logG0)

#### 2.3.3. Matsuda Index

The Matsuda Index (ISI) was induced as an index to evaluate whole-body physiological insulin sensitivity from the data obtained by OGTT [15]. This index is calculated from plasma glucose (mg/dL) and insulin (mIU/L) concentrations in the fasting state (I0,G0) and during OGTT (Imean, Gmean) [15].
ISI = 1000/√((G0 × I0) × (Gmean × Imean))

#### 2.3.4. Indices of Insulin Secretion

Beta cell function was estimated in the fasting state with [13]
HOMA-β = (20 × I0)/(G0 − 3.5),
and during the OGTT with the Stumvoll index [16]:1st phase = 1283 + 1.829 × I30 − 138.7 × G30 + 3.772 × I0 plus 2nd phase = 286 + 0.416 × I30 − 25.94 × G30 + 0.926 × I0,
and the ratio of the incremental insulin (Ins30) to glucose response (Glc30) over the first 30 min during the OGTT ΔIns30/ΔGlc30 [17].

### 2.4. Intravenous Glucose Tolerance Test (IVGTT)

After 12 h or 36 h of fasting, respectively, 0.3 g/kg bodyweight of 20% glucose solution were given at time 0 min. Blood was drawn from the contralateral antecubital vein at −10, 0, 2, 4, 6, 8, 10, 20, 30, 40, 50, 60, 80 and 100 min for the assessment of plasma glucose and insulin concentrations.

### 2.5. Biochemical Measurements

Insulin and C-peptide were measured by chemiluminescence on an Advia Centaur system (Siemens Healthcare Diagnostics, Eschborn, Germany). Plasma glucose and routine parameters were determined using a cobas analyzer (Roche Diagnostics, Mannheim, Germany).

### 2.6. Serum Sample Preparation for Metabolomics

To extract metabolites, collected serum samples tubes were treated following a previously described protocol [18]. Briefly, 50 µL were vortexed for 5 min with 500 µL of ice-cold extraction mixture (MeOH/water, 9/1, −20 °C, with a cocktail of internal standards) and then centrifuged (10 min at 15,000× *g*, 4 °C). Several fractions were split to be analysed by different Liquid and Gas chromatographies coupled with mass spectrometers (LC/MS and GC/MS).

Widely targeted analysis by GC-MS/MS was performed on a 7890 A gas chromatography (Agilent Technologies, Vienna, Austria) coupled to a QQQ (triple quadrupole) 7000 C (Agilent Technologies, Vienna, Austria). Polyamines, bile acids and short chain fatty acids analyses were performed by LC-MS/MS with a 1290 UHPLC (Ultra-High Performance Liquid Chromatography) (Agilent Technologies, Vienna, Austria) coupled to a QQQ 6470 (Agilent Technologies, Vienna, Austria). Widely pseudo-targeted analysis by UHPLC-HRAMS (Ultra-High Performance Liquid Chromatography—High Resolution and Accuracy Mass Spectrometer) was performed on a Dionex U3000/Orbitrap q-Exactive (Thermo Fisher Scientific, Waltham, MA, USA) coupling. These methods were previously described by Durand et al. 2021 [19].

### 2.7. Sample Size Calculation

Data on the effects of starvation on glucose metabolism in different populations are very limited. Therefore, for sample size calculation we used preliminary data from our InterFAST-Trial [20] which included only healthy, non-obese participants. Hence, for the current study, a mean difference in 2 h-glucose of 20 ± 25 mg/dL between 12 h and 36 h fasting was assumed for healthy participants. Based on a paired *t*-test (two-sided, alpha 5%, power 90%), 19 subjects were required for each cohort investigated to demonstrate the assumed difference. Finally, we decided to include 20 subjects with complete follow-up in each cohort.

### 2.8. Statistical Analysis

Each cohort was analysed separately. Results are presented as means (±standard deviations). Normality was assessed by means of the Kolmogorov-Smirnov test. Differences in baseline characteristics between the intervention cohorts were summarized using one-way ANOVA with a Tukey post hoc test. Intra-group differences between 12 h and 36 h of fasting were analysed either by the paired *t*-test or Wilcoxon signed-rank test as appropriate. Missing values were not replaced. Areas under the curve (AUC) were calculated for C-peptide, glucose and insulin based on the trapezoidal rule. A two-tailed *p* < 0.05 was used for statistical significance. Data were analysed in SPSS Statistics software (Version 23; IBM Corp., Armonk, New York, NY, USA). Figures were created in R software for Data Analysis and Graphics (4.0).

### 2.9. Metabolomics Data Analysis

All targeted treated data were merged and cleaned with a dedicated R (version 4.0) package (@Github/Kroemerlab/GRMeta). The metabolomics data were analyzed and visualized with Metaboanalyst 5.0 [21] or Graphpad Prism 9 for Mac OS X, GraphPad Software, San Diego, California USA. Missing metabolomics data were either replaced by 1/5 of the minimum positive value of each metabolite or by the mean of each metabolite, for 12 h (3.7%) and ratio-based (36 h/12 h values) (6.6%) analyses, respectively. One data set outside the group’s 95% confidence interval was excluded (participant 27, non-obese cohort) from the metabolomics analysis (Appendix A). Sparsed Partial Least Squares Discriminant Analysis (sPLS-DA) was performed using 25 metabolites per component. Significant differences in the 12 h metabolomes were detected by one-way ANOVAs followed by false discovery rate (FDR) correction. Metabolite classifications were generated using Metaboanalyst’s chemical structures sub-class option. Spearman’s correlation matrices were calculated with the corrplot and Hmisc packages in R, using Rstudio. Prism 9.0 was used for simple linear regression analyses between selected metabolites and OGTT-based 12 h glucose-AUC values. After testing for normality with Shapiro-Wilk tests, group comparisons of selected, normalized metabolites were performed with either one-way ANOVAs or Kruskal-Wallis tests, accounting for multiple comparisons with post-hoc Tukey or Dunn’s tests, respectively (Appendix A).

### 2.10. Primary and Secondary Outcomes

The primary outcome measure was the difference in 2-h glucose levels after 12 h versus 36 h of fasting assessed during an OGTT in the three predefined patient cohorts. Secondary outcomes included changes in glycemic patterns and insulin sensitivity indices.

## 3. Results

### 3.1. Baseline Characteristics

In total 60 participants (mean age of 43 ± 16 years, 62% women), 20 per cohort, were included of which all completed the trial. Baseline characteristics of the three study cohorts (non-obese, obese and type 2 diabetes) are shown in Table 1.

### 3.2. Glycaemic Parameters

#### 3.2.1. Non-Obese Healthy Subjects

While fasting glucose levels were lower after 36 h fasting (73 ± 11 mg/dL vs. 80 ± 7 mg/dL; *p* = 0.003), the mean 2-h glucose levels during the OGTT were significantly higher after 36 h fasting compared to 12 h (109 ± 31 mg/dL vs. 79 ± 18 mg/dL; *p* = 0.001) as was the AUC of glucose (21,627 ± 4002 mg/dL vs. 17,070 ± 3128 mg/dL; *p* < 0.001), while insulin levels at 30 min were lower after prolonged fasting (41.2 ± 34.1 mU/L vs. 56.1 ± 29.7 mU/L; *p* = 0.039) (Figure 1A,B). Lower fasting C-peptide levels were observed after 36 h fasting as compared to 12 h fasting [2.5 (1.8–4.1) vs. 4.3 (4.0–7.1) mU/L; *p* = 0.001]. In the IVGTTs, similar glucose patterns with higher excursion at 30 min, 40 min and 50 min following 36 h fasting were observed (Appendix A), with reduced insulin levels (Appendix A).

All three indices of insulin sensitivity, QUICKI, HOMA-IR and Matsuda, improved after the prolonged fasting period, 1st phase and 2nd phase insulin secretion, and the incremental insulin to glucose ratio were lower after 36 h fasting (Table 2).

#### 3.2.2. Obese Subjects

Two subjects had an HbA1c within the range of prediabetes (5.7–6.4 mmol/mol) and all the others had HbA1c readings below the prediabetes threshold. Significantly lower fasting glucose levels (88 ± 13 mg/dL vs. 94 ± 10 mg/dL; *p* = 0.002), fasting insulin levels [7.2 (5.0–14.5) mU/L vs. 11.4 (6.7–15.9) mU/L; *p* = 0.020] and C-peptide levels [1.65 (0.95–2.24) ng/mL vs. 2.00 (1.55–2.46) ng/mL; *p* = 0.036] were observed after 36 h as compared to 12 h fasting (Figure 1C,D). However, 2 h-glucose levels after OGTT did not significantly differ. While glucose values were similar after 36 h and 12 h of fasting in the OGTT, glucose levels were significantly higher after 36 h fasting in the IVGTT (20–100 min) (Appendix A), with reduced insulin levels (Appendix A). Both QUICKI and HOMA-IR improved after 36 h fasting (Table 2).

#### 3.2.3. Subjects with Type 2 Diabetes (T2D)

Four patients were on diabetes diet only, while 16 had a metformin monotherapy (Hb) or were on a combination with other oral glucose lowering drugs (4 on DPP4-inhibitors and 7 on SGLT2-inhibitors). Significantly lower fasting glucose (136 ± 26 mg/dL vs. 150 ± 37 mg/dL; *p* = 0.017), fasting insulin (9.2 ± 5.8 vs. 11.2 ± 4.4 mU/L; *p* = 0.007) and c-peptide (1.94 ± 1.02 vs. 2.31 ± 0.75; *p* = 0.002) levels were found after 36 h of fasting, while 2-h post-challenge glucose was not significantly different between the two fasting periods. All three indices of insulin sensitivity significantly improved after the prolonged fasting period. Fasting proinsulin levels were significantly lower after prolonged fasting (20.7 (10.6–33.0) pmol/L vs. 23.6 (15.3–37.1) pmol/L; *p* = 0.019). Glucose and insulin curves did not significantly differ after 12 h or 36 h fasting, neither in the OGTT nor in the IVGTT (Table 2, Figure 1E,F, Appendix A).

**Table 2 nutrients-15-00511-t002:** Effects of 12 h and 36 h fasting on parameters of glucose metabolism.

	Non-Obese Cohort (*n* = 20)			Obese Cohort (*n* = 20)			Type 2 Diabetes Cohort (*n* = 20)		
	after 12 h Fasting	after 36 h Fasting	*p*-Value	after 12 h Fasting	after 36 h Fasting	*p*-Value	after 12 h Fasting	after 36 h Fasting	*p*-Value
Fasting glucose (mg/dL)	80 ± 7	73 ± 11	0.003	94 ± 10	88 ± 13	0.002	150 ± 37	136 ± 26	0.017
Plasma glucose 120 min (mg/dL)	79 ± 18	109 ± 31	0.001	108 ± 36	117 ± 29	0.067	262 ± 83	282 ± 59	0.121
Fasting insulin (mU/L)	4.3 (4.0–7.1)	2.5 (1.8–4.1)	0.002	11.4 (6.7–15.9)	7.2 (5.0–14.5)	0.020	11.2 ± 4.4	9.2 ± 5.8	0.007
Fasting C-peptide (ng/mL)	0.94 (0.83–1.13)	0.61 (0.43–0.82)	0.001	2.00 (1.55–2.46)	1.65 (0.95–2.24)	0.036	2.31 ± 0.75	1.94 ± 1.02	0.002
Glucose (mg/dL) AUC (in minutes)	17,070 ± 3128	21,627 ± 4002	0.000	22,638 ± 3872	23,315 ± 3622	0.197	46,771 ± 11532	46,238 ± 7822	0.734
Insulin (mU/L) AUC (in minutes)	5592 ± 2679	7557 ± 5448	0.137	12,621 ± 8364	13,794 ± 9254	0.629	7695 ± 4427	8337 ± 4394	0.392
QUICKI	0.39 (0.37–0.40)	0.45 (0.41–0.49)	0.000	0.33 (0.31–0.35)	0.36 (0.32–0.39)	0.021	0.31 (0.30–0.33)	0.32 (0.31–0.37)	0.001
HOMA_IR_	0.86 (0.75–1.41)	0.43 (0.27–0.67)	0.001	2.52 (1.72–4.27)	1.37 (0.90–3.07)	0.001	4.26 ± 2.06	3.17 ± 2.17	0.012
ISI	1.28 ± 0.95	1.88 ± 1.34	0.004	0.40 (0.32–0.53)	0.54 (0.27–0.72)	0.078	0.26 (0.22–0.48)	0.30 (0.20–0.68)	0.014
Proinsulin (pmol/L)	3.9 (3.4–5.0)	3.6 (2.7–4.6)	0.111	9.9 (7.0–25.9)	11.1 (6.6–16.6)	0.151	23.6 (15.3–37.1)	20.7 (10.6–33.0)	0.019
1st phase insulin secretion	1138 ± 496	742 ± 471	0.001	1400 ± 956	1408 ± 759	0.972	−68 ± 635	−17 ± 585	0.536
2nd phase insulin secretion	295 ± 108	211 ± 104	0.001	370 ± 224	370 ± 176	0.994	68 ± 130	73 ± 125	0.740
HOMA-β	105.4 (75.9–161.7)	63.6 (18.4–175.9)	0.202	140.3 (89.1–240.0)	149.1 (82.3–252.9)	0.963	53.2 ± 31.8	47.1 ± 32.4	0.102
ΔIns_30_/ΔGlc_30_	1.14 (0.95–1.50)	0.49 (0.13–0.86)	0.001	0.83 (0.50–1.91)	0.92 (0.47–2.11)	0.762	0.23 ± 0.23	0.21 ± 0.22	0.157

AUC: area under the curve; QUICKI: quantitative insulin sensitivity index; HOMA_IR_: homeostatic model assessment for insulin resistance; HOMA-β: homeostatic model assessment for beta cell function; ISI: Matsuda insulin sensitivity index.

#### 3.2.4. Comparison between Groups

When we performed a non-prespecified comparison of the delta of 2 h glucose between 12 and 36 h of fasting, adjusted for fasting glucose, a significant difference between the participant groups (non-obese, obese, type 2 diabetes) was found (*p* = 0.001 using ANOVA). Likewise, we found a significant difference between the groups when comparing AUC for glucose during the oGTT (*p* = 0.005, using ANOVA).

### 3.3. Metabolomics Results

#### 3.3.1. Twelve Hours Fasting Period

We identified 195 metabolites by widely targeted mass spectrometry in the serum metabolomes (Figure 2A, Appendix A). Sparsed Partial Least Squares Discriminant Analysis (sPLS-DA) revealed relatively small global differences in the 12 h-metabolomes (Appendix A). The top discriminating metabolites included sebacic acid (decanedioic acid), histidine, glucose, inositol and gamma-glutamyl-tyrosine (Appendix A). Besides glucose, sixteen metabolites were considered significantly different between the cohorts (ANOVA FDR-corrected *p* < 0.1) (Appendix A), which showed evident clusters in a heatmap of the 50 top ANOVA-ranked metabolites (Figure 2B). These metabolites could be grouped into classes known to be affected by obesity and diabetes, including amino acids, fatty acyl carnitines and fatty acids, among others (Figure 2C).

Besides glucose (Figure 2D), other metabolites that were significantly elevated in people with type 2 diabetes included 2-hydroxybutyrate, carnitine C3:0, carnitine C8:1, cystine, dodecanedioic acid, histidine, inositol, sebacic acid, threonic acid and tyrosine, while glycylglycine was significantly decreased (Appendix A), suggesting disturbances in lipid and amino acid metabolism. The obese cohort shared some of these alterations, including elevated carnitines C3:0 and C8:1 and reduced glycylglycine (Appendix A). Alterations exclusively identified in the obese cohort included reduced arachidic acid, homoserine and indole-3-aldehyde, as well as increased gamma-glutamyl-tyrosine (Appendix A).

Across all cohorts, four metabolites were strongly correlated with the OGTT-based glucose-AUC after the 12 h overnight fast (Figure 2E,F). Cohort-specific analyses then revealed significant positive correlations of these metabolites with the glucose-AUC in at least one of the cohorts (*p*-values for Non-obese/Obese/Type 2 Diabetes for 2-hydroxybutyric acid: 0.030/0.011/0.298; histidine: 0.029/0.179/<0.001; sebacic acid: 0.041/0.219/<0.001; tyrosine: 0.026/0.491/<0.001).

#### 3.3.2. Thirty-Six Hours Fasting Period

To identify fasting-responsive metabolites we looked for metabolites that significantly changed within each cohort (0.8 < FC > 1.2, FDR-corrected *p*-value <0.1) and identified 37, 19 and 10 metabolites in the non-obese, obese and type 2 diabetes cohort, respectively (Figure 3A and Appendix A), all belonging to classes known to be affected by fasting in mammals, including ketone bodies, fatty acids, polyunsaturated fatty acids, carnitines and amino acids (Figure 3B). Seven of these metabolites were shared across all three cohorts according to the selected thresholds (Figure 3C,D). The metabolic response to fasting appeared attenuated in obese and diabetic people (Figure 3E), including reduced increases of the ketone bodies and 2-hydroxybutyrate (n.s. and *p* < 0.05, respectively) (Figure 3F).

## 4. Discussion

Our study demonstrates that the insulin secretory response and consequently the glucose variations following oral or intravenous glucose tolerance tests after a 36 h fasting period differ among non-obese, obese and people with established type 2 diabetes. While we observed a reduced early insulin response in non-obese, healthy subjects after 36 h of fasting as compared to 12 h, this finding persisted in obese people only in the IVGTT and disappeared completely in those with type 2 diabetes. However, fasting insulin levels were lower and insulin sensitivity improved after 36 h in all three cohorts.

Our data confirm previous studies, which demonstrated a reduced first-phase insulin secretion with increased post-challenge glycemic excursions [7,8,22,23]. Recently, JØrgensen et al. showed a significant difference in organ-specific insulin action after 36 h of fasting in healthy young men. While peripheral insulin sensitivity declined, hepatic insulin sensitivity was significantly increased following prolonged fasting. In this context, reduced insulin secretion may be interpreted as a physiologic response to the improved insulin action allowing the beta-cell to rest [22]. Hence, the beta-cell physiologically reduces insulin secretion in the fasting state to avoid hypoglycemic events [24]. Goginashvili and colleagues demonstrated that starved beta-cells reduce insulin release by suppressing autophagy, hence differing from most mammalian cell types that respond to starvation by increasing autophagic flux [25]. However, with increasing insulin resistance and finally overt diabetes, resulting in impaired beta-cell function [24], metabolic adaptions observed in healthy subjects decline, as demonstrated by our data. Moreover, previous studies have shown that the incretin effect, which leads to an insulin secretion from the beta-cell following oral carbohydrate administration, is also impaired in people with obesity and ultimately in those with type 2 diabetes [26]. While the oral glucose tolerance test leads to a blood glucose induced and incretin induced stimulation of the insulin secretion from the pancreas, the IVGTT stimulates insulin secretion by the rapid increase in blood glucose only [27]. Although the incretin effect is known to be reduced in people with obesity [26], no data on the effects of prolonged fasting on incretin release are available. However, we did not perform an iso-glycemic intravenous glucose infusion, matching the OGTT glucose curve, to definitively investigate the incretin effect in our study.

It is assumed, that hunger is a major limiting factor for the success of restrictive diets [28]. Sundfør and colleagues showed in a one year intervention trial that hunger levels were higher in those following intermittent fasting compared to continuous energy restriction (CER) [28], while several other trials showed no difference in hunger or satiety between those following IF compared to CER [29,30].

In our trial, information about hunger and satiety was not recorded using a standardized questionnaire given that we only investigated one single episode of 36 h fasting.

A decline in resting metabolic rate (RMR) is described for most caloric restriction interventions [31,32]; however, interestingly we did not observe a significant change in RMR in people following an intermittent fasting regimen, neither in healthy, normal-weight people [20] nor patients with type 2 diabetes [33]. Dietary interventions without reducing RMR are crucial, in particular with regard to weight regain following the intervention.

While metabolomics profiling was previously shown to be capable of detecting early type 2 diabetes and of helping to elucidate the underlying pathophysiology [34], we used metabolomic analysis to understand the spectrum of metabolic changes associated with prolonged fasting. The 12 h metabolome showed typical alterations associated with obesity and type 2 diabetes, including increases in acyl-carnitines, as has been shown previously in obese adults and children [35,36]. Likewise, LaBarre et al. recently demonstrated in overweight and obese adolescents a blunted decline of acyl-carnitines and fatty acid oxidative intermediates as a response to OGTT [37]. The association analysis of the AUC for glucose during the OGTT in our study suggests 2-hydroxybutyric acid, histidine, sebacic acid and tyrosine as possible biomarkers for disturbed glucose metabolism under standard overnight-fasted conditions. Interestingly, 2-hydroxybutyric acid and sebacic acid have been suggested as markers for insulin resistance [38,39,40], which is corroborated by their elevated levels in our diabetic cohort (Appendix A). Conversely, both sebacic acid and histidine are investigated as dietary supplements to improve glycemic control in diabetic patients, which however conflicts with our findings as sebacic acid levels directly correlate with glucose levels [41,42]. Moreover, contradictory results concerning tyrosine and histidine levels in diabetes and metabolic syndrome have been reported [43,44,45,46]. However, it remains unknown which of the identified group differences or changes in metabolite levels during fasting directly contribute to the pathologies of obesity or type 2 diabetes. Glucose-lowering medications in type 2 diabetes patients further complicate the interpretation of the metabolomics data, as their use was shown to alter human metabolite profiles [47].

We were further interested in the metabolic changes observed after prolonged fasting. Similar to the OGTT and IVGTT investigations, the overall pattern of the metabolic fasting response appeared attenuated in obese and people with diabetes, suggesting a systemically reduced metabolic flexibility in these patients following prolonged fasting (Figure 3).

## 5. Conclusions

Our study is afflicted by several limitations. Firstly, the mean age differed significantly between the three cohorts, with those having type 2 diabetes being the oldest. This age difference may impact the insulin and glucose responses to OGTT and IVGTT, hence constituting a possible confounding factor, also for the findings in the metabolomics patterns. Second, the glucose-lowering treatment used for the management of type 2 diabetes might have influenced the OGTT and IVGTT response. However, we aimed to minimize this source of bias by pausing the glucose-lowering medications on the day of investigation. Third, the objective of the study was to compare the impact of 12 and 36 h fasting within the respective groups of non-obese, obese and people with type 2 diabetes. Although we did perform a comparison of the delta of 2 h glucose and the AUC of glucose between the 12 and 36 h fasting period across the groups and identified significant differences, those results need to be interpreted with caution, as they were not prespecified and the study sample was not sufficiently powered for those analyses.

The major strength of our study is the direct comparison of 36 h versus 12 h fasting periods, the glucometabolic adaptions and flexibility of the metabolism in the insulin sensitivity spectrum reaching from healthy, normal weight to obese subjects and, finally, people with established type 2 diabetes. Our OGTT data are supported by IVGTT tests and suggest that the incretin effect might be altered following prolonged fasting in obese subjects, which will need to be investigated further.

## Figures and Tables

**Figure 1 nutrients-15-00511-f001:**
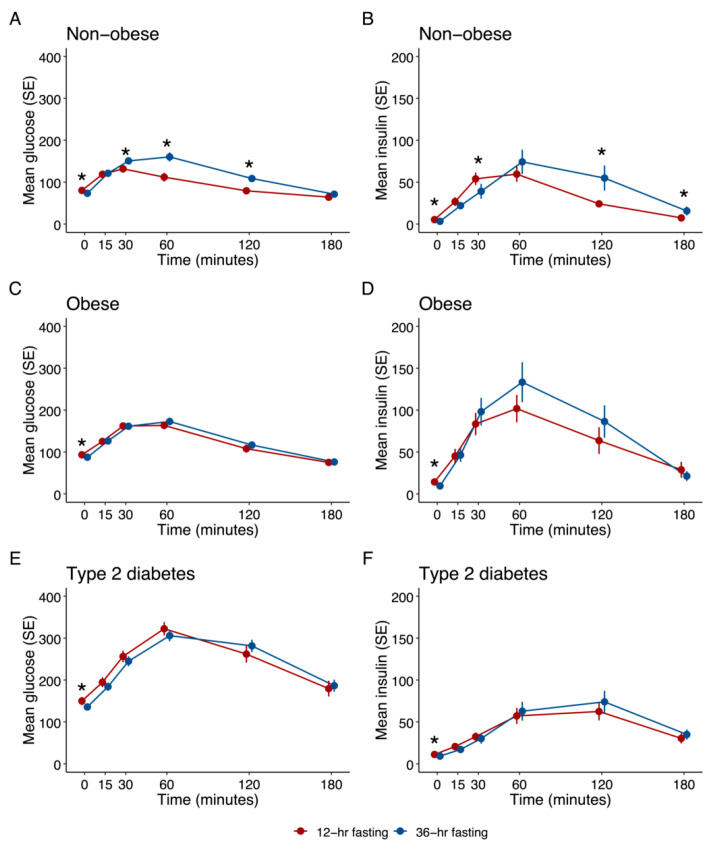
Effects of 12 h and 36 h fasting on glucose metabolism during OGTT (n = 20 in each cohort); * *p* < 0.05. (**A**) Effect on plasma glucose in non-obese cohort (**B**) Effect on serum insulin in non-obese cohort (**C**) Effect on plasma glucose in obese cohort (**D**) Effect on serum insulin in obese cohort (**E**) Effect on plasma glucose in type 2 diabetes cohort (**F**) Effect on serum insulin in type 2 diabetes cohort.

**Figure 2 nutrients-15-00511-f002:**
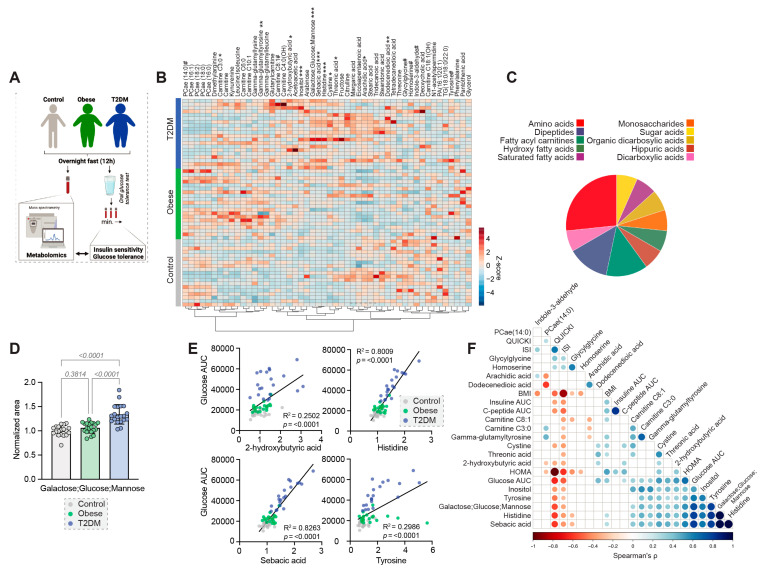
Differences in the 12 h metabolomes. (**A**) Schematic overview of the analysis approach. ((**A**) was created with BioRender.com) (**B**) Heatmap of the 50 top ANOVA-ranked metabolites. FDR-corrected *p*-values # < 0.1, * < 0.05, ** < 0.01, *** < 0.001. (**C**) Classification of significantly different metabolites. (**D**) Differences in the levels of Galactose; Glucose; Mannose, normalized to the mean of the non-obese cohort. (**E**) Simple linear regression analyses of selected metabolites and the glucose AUC during OGTT. (**F**) Spearman correlation matrix between significantly different metabolites and glycaemic parameters. The color and size of the dots correspond to the correlation coefficient. *p*-value cut-off 0.05.

**Figure 3 nutrients-15-00511-f003:**
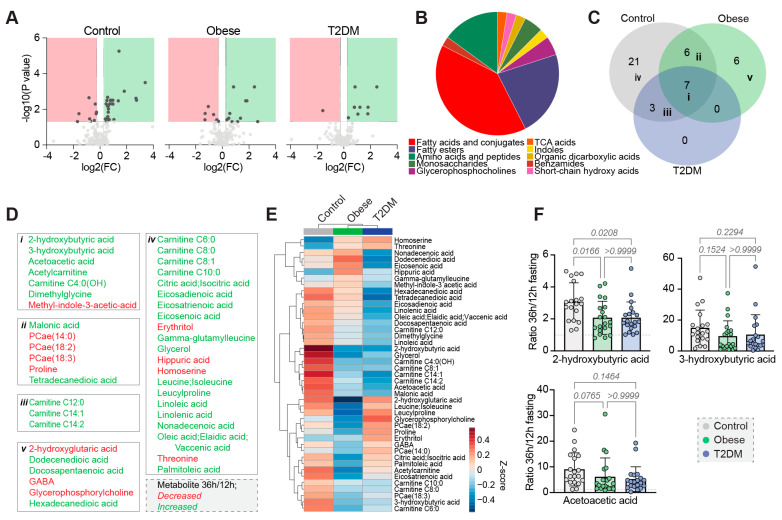
Differences in the metabolomes after prolonged fasting. (**A**) Volcano plots of fastinginduced changes in the metabolome. FC = fold change. (**B**) Classification of significantly different metabolites. (**C**) Venn diagram of the identified fasting-responsive metabolites. (**D**) All fasting-responsive metabolites allocated to the groups of the Venn diagram. (**E**) Heatmap of the fasting-responsive metabolites, showing cohort averages. (**F**) Group comparisons of the ratios of selected metabolites.

**Table 1 nutrients-15-00511-t001:** Baseline characteristics of non-obese and obese participants and patients with type 2 diabetes.

	Non-Obese Cohort (*n* = 20)	Obese Cohort (*n* = 20)	Type 2 Diabetes Cohort (*n* = 20)
**Age (years)**	32 ± 10	37 ± 10	60 ± 9
**Bodyweight (kg)**	67.5 ± 9.6	102.7 ± 14.1	64.9 ± 23.9
**Height (cm)**	173 ± 10	174 ± 8	176 ± 8
**BMI (kg/m^2^) ^1^**	22.6 ± 1.7	34.6 ± 4.8	31.0 ± 7.1
**Blood pressure systolic (mmHg)**	117 ± 11	122 ± 10	123 ± 12
**Blood pressure diastolic (mmHg)**	76 ± 13	79 ± 11	74 ± 8
**Waist to hip ratio**	0.8 ± 0.1	0.9 ± 0.1	1.0 ± 0.1
**Fasting glucose (mg/dL)**	80.3 ± 7.3	93.5 ± 14.9	149.9 ± 36.9
**Fasting insulin (mU/L)**	5.3 ± 2.5	14.3 ± 14.9	11.1 ± 4.3
**HbA1c (mmol/mol)**	32 ± 2	35 ± 3	53 ± 8

^1^ BMI: body mass index.

## Data Availability

The datasets generated and/or analyzed during the current study are available from the corresponding author.

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
