# Peer review of "Glucose Metabolism and Metabolomic Changes in Response to Prolonged Fasting in Individuals with Obesity, Type 2 Diabetes and Non-Obese People—A Cohort Trial"

_nutrients, 2023, doi:10.3390/nu15030511_

Round 1

Reviewer 1 Report

The authors offer an interesting prospective mechanistic parallel cohort trial observing differences in OGTT and IVGTT at 12 hours and 36 hours as the primary outcome between three groups: non-obese, obese (non-diabetic), and patients diagnosed with type two diabetes. Additionally, the authors utilized high throughput metabolomic analysis to determine baseline differences within the three groups at the specified time points. 

There are numerous strengths to the study. The authors strictly controlled the timepoints the three groups were assessed at 12 hours and 36 hours. These are interesting and appropriate time points as they are reasonable times to assess for the initiation of metabolic changes noted in fasting, but prior to the body entering starvation. Their appreciation for evaluating not just OGTT, but also performing stringent calculations to observe insulin resistance and sensitivity ( HOMA-IR, QUICKI, and Matsuda Index) was a thorough and thoughtful analysis. 

The authors note that the study is not powered to support the metabolomic comparisons between the groups, however the notable differences between the three groups, particularly with regard to the discriminating metabolites of sebacic acid, histidine, inositol and gamma-glutamyltyrosine. The differences in metabolites noted in the obese and the T2DM groups offer discovery information that can provide future biochemical markers to diagnose early diabetes. 

This study is very timely and interesting, but there are several weaknesses. The authors note that the T2DM group are notably older than the other two groups. Is this an effect of an "old" burned out pancreas, early malignancy (no noted long term follow up), or true metabolic dysfunction.

Not all T2DM are metabolically equal. Were these patients obese or not? It is interesting that the mean bodyweight is 64.9 with a large SD and the W:H ratio is 1.0, notably large than both groups, including the obese group. There is a growing number of "skinny" diabetics who are phenotypically not obese, but metabolically unfit. Just as the metabolomic data suggests, perhaps many of the "obese" cohort is pre-diabetic. It would have been prudent to include HgbA1C in the measurements. The authors note that 16 of the 20 T2DM subjects were on "metformin or combination therapy". There should be more detail about this. What doses? Which combo therapies? This makes a real difference, particularly with regard to the metabolomic data. It would have been more interesting to include 20 T2DM patients who are diet controlled without pharmacologic manipulation of their glucose and insulin utilization and production. 

The study was powered based on a prior study and a t-test. This is for normally distributed data, which the raw data in such a study is probably not normally distributed. Even if the study is appropriately powered for 19 in each cohort, accruing only 20 and expecting completion in the three groups was bold. 

Finally, I believe the OGTT and IVGTT data imploring further questioning of the role of incretin is fascinating and novel work. I encourage the authors to pursue this further. It is a bold and erroneous statement to say you "have unravelled" the metabolomic shifts induced by prolonged fasting. The study wasn't powered to support this and you have only reached the tip of the iceburg. 

Author Response

The authors offer an interesting prospective mechanistic parallel cohort trial observing differences in OGTT and IVGTT at 12 hours and 36 hours as the primary outcome between three groups: non-obese, obese (non-diabetic), and patients diagnosed with type two diabetes. Additionally, the authors utilized high throughput metabolomic analysis to determine baseline differences within the three groups at the specified time points. 

There are numerous strengths to the study. The authors strictly controlled the timepoints the three groups were assessed at 12 hours and 36 hours. These are interesting and appropriate time points as they are reasonable times to assess for the initiation of metabolic changes noted in fasting, but prior to the body entering starvation. Their appreciation for evaluating not just OGTT, but also performing stringent calculations to observe insulin resistance and sensitivity (HOMA-IR, QUICKI, and Matsuda Index) was a thorough and thoughtful analysis. 

A: We would like to thank the reviewer for appreciating our study design.

The authors note that the study is not powered to support the metabolomic comparisons between the groups, however the notable differences between the three groups, particularly with regard to the discriminating metabolites of sebacic acid, histidine, inositol and gamma-glutamyltyrosine. The differences in metabolites noted in the obese and the T2DM groups offer discovery information that can provide future biochemical markers to diagnose early diabetes. 

A: We would like to thank the reviewer for pointing this out. We agree that our study suggests potential novel biomarkers and important information on metabolic changes. However, for full transparency we also wanted to highlight those limitations of this study.

This study is very timely and interesting, but there are several weaknesses. The authors note that the T2DM group are notably older than the other two groups. Is this an effect of an "old" burned out pancreas, early malignancy (no noted long term follow up), or true metabolic dysfunction.

A: Thank you for raising this issue. The mean C-peptide level in the T2DM group after 12 hours of fasting was 2.31 ± 0.75 ng/ml, suggesting a reasonable remaining insulin secretion in this group, however, given a mean fasting glucose level of 262 ±83 mg/dl, the compensatory insulin secretion capacity is severely reduced. This is also reflected by the other measures of beta-cell function and insulin sensitivity. Overall, we believe that our study cohort displays insulin resistance together with significant beta-cell dysfunction, as one would expect from a typical T2DM cohort.
As the study participants participate in our long-term diabetes registry project, we can tell, that 12 months after the end of the study, only one case of breast cancer occurred in the T2DM cohort.

Not all T2DM are metabolically equal. Were these patients obese or not? It is interesting that the mean bodyweight is 64.9 with a large SD and the W:H ratio is 1.0, notably large than both groups, including the obese group. There is a growing number of "skinny" diabetics who are phenotypically not obese, but metabolically unfit. Just as the metabolomic data suggests, perhaps many of the "obese" cohort is pre-diabetic. It would have been prudent to include HgbA1C in the measurements. The authors note that 16 of the 20 T2DM subjects were on "metformin or combination therapy". There should be more detail about this. What doses? Which combo therapies? This makes a real difference, particularly with regard to the metabolomic data. It would have been more interesting to include 20 T2DM patients who are diet controlled without pharmacologic manipulation of their glucose and insulin utilization and production. 

A: We have now included all the glucose lowering drugs used in the T2DM group (4 people were on a DPP-4 inhibitor, 7 on an SGLT2 inhibitor). The mean metformin dose was 1210 ± 520 mg/day. This information was now added to the manuscript. We do also agree that the mean HbA1c values are important and have now added them to table 1. In the obese group 2 subjects were within the prediabetic range.

The study was powered based on a prior study and a t-test. This is for normally distributed data, which the raw data in such a study is probably not normally distributed. Even if the study is appropriately powered for 19 in each cohort, accruing only 20 and expecting completion in the three groups was bold. 

A: As we had no further supporting data available when designing the study, we had to rely on our previous study. However, the chosen wording regarding the enrolment of the study participants was not correct in the manuscript – we planned to enrol 20 participants in each group who completed all study visits. Dropouts would have been replaced, however, no participant dropped out, probably due to the short study duration. We have revised the text.

Finally, I believe the OGTT and IVGTT data imploring further questioning of the role of incretin is fascinating and novel work. I encourage the authors to pursue this further. It is a bold and erroneous statement to say you "have unravelled" the metabolomic shifts induced by prolonged fasting. The study wasn't powered to support this and you have only reached the tip of the iceburg. 

A: We agree that the statement is overemphasizing our results. We have removed this last sentence of the discussion. Further studies looking into the effects of fasting on the incretin effects are currently in discussion.

Reviewer 2 Report

The manuscript is new and the authors study an aspect of growing interest. The methodology used is adequate and the results support the discussion. However, I have the following comments.

I. Major comments:

1. In the introduction I suggest including a brief paragraph on the neurological aspects involved in prolonged fasting.

2. The discussion is good. However it is necessary to discuss:

2.1. Aspects related to regulation of metabolism, especially energy

2.2. Appetite and satiety

3. The manuscript is well written, but it requires correcting editorial errors.

4. Some important ideas (especially in the discussion) required references. For example, the paragraph "The

nature of these differences .......... following prolonged fasting". Discussion section, line 91 to line 98.

5. Improve the conclusion. It is not easy to understand.

II. Minor comments:

1. Improve the wording of the study objective

2. Improve the description of the type of study (methodology section - study design)

3. The figures are very good, but I suggest increasing the size of the figures, especially the letters.

4. It is necessary to correct some editorial errors.

Author Response

The manuscript is new and the authors study an aspect of growing interest. The methodology used is adequate and the results support the discussion. However, I have the following comments.

  1. Major comments:
  2. In the introduction I suggest including a brief paragraph on the neurological aspects involved in prolonged fasting.

      A: Many thanks for this suggestion. We added a brief paragraph about the                  effects of prolonged fasting on cognitive function to the introduction section.

  1. The discussion is good. However it is necessary to discuss:

     2.1. Aspects related to regulation of metabolism, especially energy

          A: This is indeed an important topic. We do now discuss the data available                  on the impact of caloric restriction / intermittent fasting on energy                         expenditure.

      2.2. Appetite and satiety

     A: We agree that these are important topics, although the data are somewhat             ambiguous. We have added a paragraph in the main manuscript.

  1. The manuscript is well written, but it requires correcting editorial errors.

         A: We have thoroughly revised the entire manuscript.

  1. Some important ideas (especially in the discussion) required references. For example, the paragraph "The nature of these differences .......... following prolonged fasting". Discussion section, line 91 to line 98.

        A: Many thanks for pointing this out. During the revision process, this                          particular sentence was rephrased into a more hypothetical wording.                     However, we have added several new references during the revision in the               discussion section. 

  1. Improve the conclusion. It is not easy to understand.

          A: We have rephrased the conclusion section.

 Minor comments:

  1. Improve the wording of the study objective

        A: Thank you for making us aware, that the sentence was long and difficult                to follow. We have revised the wording.

  1. Improve the description of the type of study (methodology section - study design)

         A: We have revised the study design section to make it easier to follow.

  1. The figures are very good, but I suggest increasing the size of the figures, especially the letters.

         A: The font size has been increased according to the suggestion. We believe               that figures are now more readable.

  1. It is necessary to correct some editorial errors.

         A: We have thoroughly revised the entire manuscript.

Round 2

Reviewer 2 Report

Authors answered all my comments. Therefore, manuscript can be accepted.